# Second harmonic generation imaging reveals entanglement of collagen fibers in the elephant trunk skin dermis
Andrew K. Schulz [1], Magdalena Plotczyk [2,7], Sophia Sordilla [3,7], David C. A. Gaboriau [4], Madeline Boyle [1], Krishma Singal [5], Joy S. Reidenberg [6], David L. Hu [1,3,8] ✉ & Claire A. Higgins [2,8] ✉

Form-function relationships often have tradeoffs: if a material is tough, it is often inflexible, and vice versa. This is particularly relevant for the elephant trunk, where the skin should be protective yet elastic. To investigate how this is achieved, we used classical histochemical staining and second harmonic generation microscopy to describe the morphology and composition of elephant trunk skin. We report structure at the macro and micro scales, from the thickness of the dermis to the interaction of $10\,\mu m$ thick collagen fibers. We analyzed several sites along the length of the trunk to compare and contrast the dorsal-ventral and proximal-distal skin morphologies and compositions. We find the dorsal skin of the elephant trunk can have keratin armor layers over 2 mm thick, which is nearly 100 times the thickness of the equivalent layer in human skin. We also found that the structural support layer (the dermis) of the elephant trunk contains a distribution of collagen-I (COL1) fibers in both perpendicular and parallel arrangement. The bimodal distribution of collagen is seen across all portions of the trunk, and is dissimilar from that of human skin where one orientation dominates within a body site. We hypothesize that this distribution of COL1 in the elephant trunk allows both flexibility and load-bearing capabilities. Additionally, when viewing individual fiber interactions of $10\,\mu m$ thick collagen, we find the fiber crossings per unit volume are five times more common than in human skin, suggesting that the fibers are entangled. We surmise that these intriguing structures permit both flexibility and strength in the elephant trunk. The complex nature of the elephant skin may inspire the design of materials that can combine strength and flexibility.

Elephant trunks, octopus arms, and mammalian tongues are the three canonical examples of muscular hydrostats[1]. The elephant trunk, the subject of this work, is extremely flexible and can extend by up to 25% in a telescopic manner allowing the elephant to reach distant objects[2]. The ventral side of the trunk contains oblique muscles that allow the trunk to wrap around and grasp objects[1]. It follows that the ventral surface (or posterior, caudal surface) of the trunk is often the primary point of contact with the substrate during object manipulation[3]. The dorsal surface (or anterior, rostral surface) of the trunk is not often utilized for grasping and is more exposed to external mechanical forces and predators, potentially necessitating a more protective armor-like structure. To fulfill the different roles required of it, the skin on the elephant trunk should be flexible and tough at the same time.

Relatively little work has been conducted to document the anatomy of elephant skin. In 1970, Spearman published a study discussing elephant skin's basic anatomy, including insights into the different vibrissal hairs on the trunk[4]. More recently, biomechanical studies have made connections between the skin properties and an elephant's ability to grasp and wrap its

[1]School of Mechanical Engineering, Georgia Institute of Technology, Atlanta, GA, 30332, USA. [2]Department of Bioengineering, Imperial College London, South Kensington, London, SW7 2AZ, UK. [3]School of Biological Sciences, Georgia Institute of Technology, Atlanta, GA, 30332, USA. [4]Facility for Imaging by Light Microscopy, National Heart and Lung Institute, Imperial College London, South Kensington, London, SW7 2AZ, UK. [5]School of Physics, Georgia Institute of Technology, Atlanta, GA, 30332, USA. [6]Center for Anatomy and Functional Morphology, Icahn School of Medicine at Mount Sinai, New York, NY, 10029-6574, USA. [7]These authors contributed equally: Magdalena Plotczyk, Sophia Sordilla. [8]These authors jointly supervised this work: David L. Hu, Claire A. Higgins. ✉e-mail: hu@me.gatech.edu; c.higgins@imperial.ac.uk

trunk around various objects, including barbells[3,5]. While the skin on the elephant's body is cracked for thermoregulation[6], the trunk, in contrast, has wrinkles and folds on its ventral and dorsal surfaces, respectively[5]. The skin also varies with position along the length of the trunk: the distal trunk skin (on both ventral and dorsal surfaces) is characterized by wrinkles, while the proximal dorsal trunk skin has folds. These differing skin characteristics enable the trunk to extend to reach faraway objects, with the dorsal surface stretching more than the ventral[2].

In this work, we used both classical and advanced microscopy techniques to investigate the structure of elephant trunk skin. We focused our analysis on collagen, a foundational protein that governs the structure of many body tissues, including muscle, blood vessels, and skin, and provides bio-inspiration across scales[7]. Collagen-I (COL1) is the primary collagen found within the skin; it has a fibrillar structure and can, therefore, be detected with second-harmonic generation (SHG) imaging. SHG is a nonlinear optical imaging technique that selectively detects non-centrosymmetric molecules, such as collagen, with no labeling[8,9].

SHG microscopy works by imaging the skin sample at a specific wavelength that excites the fibrillar structure of COL1; the resulting image fluorescence is roughly half the excitation wavelength coming from the multi-photon laser, hence the term "second harmonic." The fibrillar structure of COL1 fibers allows the microscopy technique to detect COL1 in the tissue, while the resulting image is related to the amount of pretension on the COL1 fibers[10], or the thickness of the fibers[11], both of which can impact the COL1 fiber's ability to resist deformation[12].

The SHG technique is label-free and, therefore, accrues less error compared to traditional histochemistry since there is not a chained sequence of staining that can vary based on the specific timing that segmented skin spends in various chemical baths[13]. The SHG technique is specific to collagen and does not pick up the other fiber structures, such as elastin or keratin, that are present within the skin[8].

In skin, COL1 networks are characterized by variations in fiber orientation, thickness, density, strain, and weaving with neighboring fibers —this last feature is a phenomenon known as entanglement[14]. These changes contribute to different tissue mechanics, such as altered elasticity. Analysis of SHG images of the skin using texture analysis techniques, such as autocorrelation or 2D Fourier transform[15], allows quantification of all these variations in COL1 fibers. SHG can further be enhanced by the application

of polarization-sensitive SHG (PSHG), where light polarized in several different curves (e.g., linear) or shapes (e.g., circular or ellipsoidal) enables the collection of images which can be analysed to reveal information on the molecular structure of collagen[16]. A form of PSHG, known as polarization-in SHG (PiSHG), has been used for diagnostic purposes, distinguishing tissue structure in diseased and cancerous tissues, compared to healthy surrounding tissue[17,18].

SHG is just one technique to understand collagen morphology and composition. Alternative approaches for imaging COL1 include polarized light microscopy (PLM), optical coherence tomography (OCT), ultrasound, and magnetic resonance imaging (MRI)[19]. These different imaging modalities and techniques allow viewing of COL1 at varying length scales, from individual fibers with PLM[20], to density and gross tissue properties with MRI[21].

Here, we used SHG to analyze COL1 architecture in elephant trunk skin. We conducted morphological and compositional analyses on skin samples along the trunk at several locations, including seven sites for SHG microscopy and eight for histochemical staining (Fig. 1). We show key differences in collagen architecture along the length of the elephant trunk, and differences between COL1 architecture in elephant and human skin.

## Results

### Macrostructure of the elephant trunk skin

The outermost layer of the skin, the stratum corneum (SC), is composed of denucleated, keratinized epithelial cells with lipids in between. Underneath the SC is the viable epidermis (VE) which is a sheet of epithelial cells with tight junctions, which gives the skin its barrier function. Beneath this is the structural support layer for the overlying epithelium, known as the dermis (D)[22]. To quantify differences in elephant skin morphology across trunk locations, we analysed images of skin sections stained with H&E to segment the skin into the SC, the VE, and D (Fig. 2 and Supplementary Fig. 1). Below, we will make comparisons of dorsal and ventral skin at the same distance from the tip of the trunk.

Starting with the stratum corneum, we found that on the dorsal trunk, the SC was thickest in the proximal base, with a mean thickness of 2 mm (Table 1 and Fig. 3A), which is significantly different from the ventral SC, with a thickness of 0.34 mm ($p < 0.001$). The remainder of the SC on the dorsal trunk varied from 0.25 to 1 mm on average (Fig. 3A). In

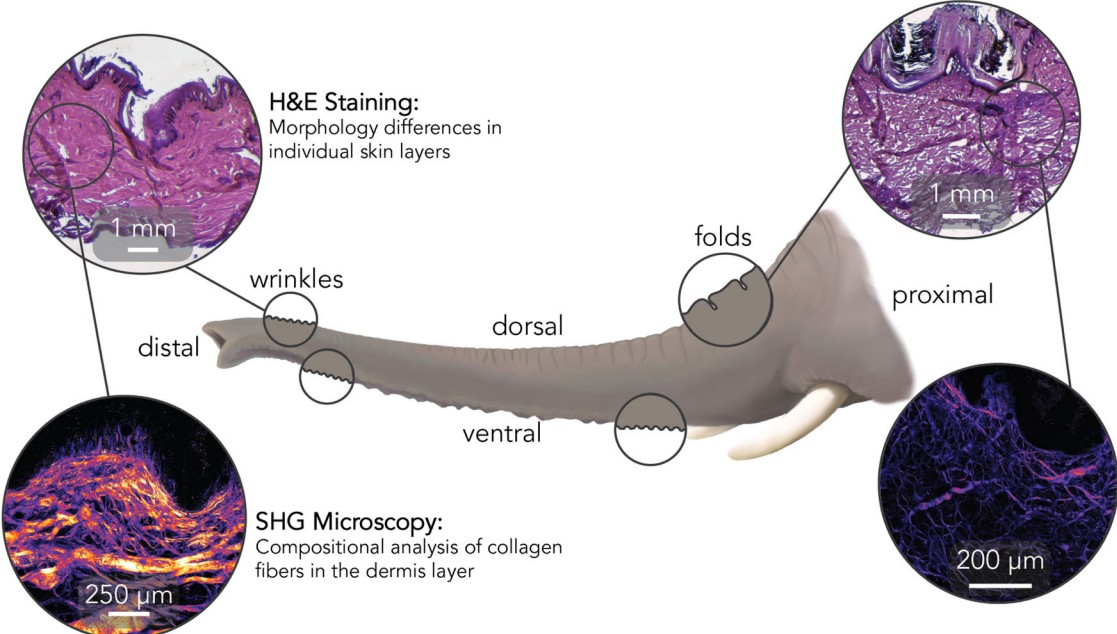

**Fig. 1 | Schematic of the elephant trunk with experimental outputs.** The top images display H&E staining along the dorsal length of the trunk with bottom images displaying SHG outputs for compositional analysis. Elephant trunk illustration by B. Seleb.

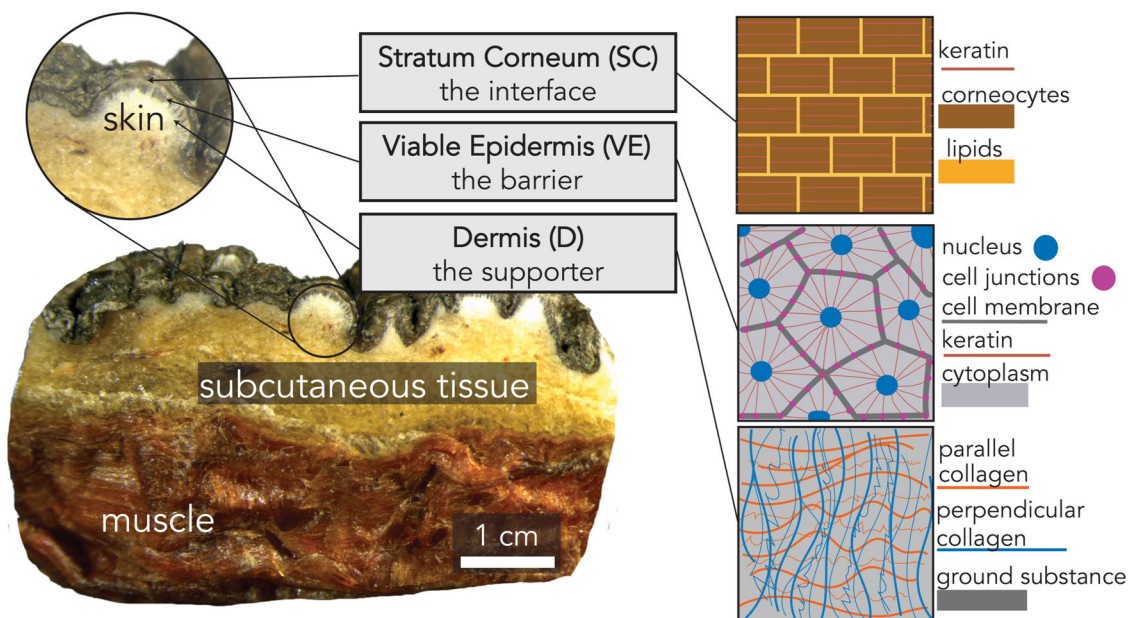

**Fig. 2 | Macroscopic image of a cross-section of elephant skin showing subcutaneous tissue and muscle.** The skin layers are shown in a schematic of the stratum corneum (SC), viable epidermis (VE), and dermis (D).

contrast, the SC of the ventral trunk had a relatively constant thickness of 0.40 mm.

The viable epidermis thickness remained broadly consistent throughout the length of the trunk and between ventral and dorsal sites (Table 1 and Supplementary Fig. 2). The overall thickness of the VE remained nearly constant at around 0.3–0.4 mm for both the dorsal and ventral elephant surfaces. An exception was the very distal tip of the dorsal skin, 3 cm from the tip (finger at the tip of the trunk), which had a tiny layer of VE at only 0.05 mm thick (Supplementary Fig. 2). This thickness displayed a statistically significant difference from the rest of the skin analyzed ($p < 0.001$).

Together, the SC and VE are considered to be the armor for the skin as they serve as the first layers of protection against environmental insults. Compared to other species' armor layers, such as scales or shells, the elephant skin on the dorsal trunk reaches 2.2 mm thick—this is double the thickness of a pangolin scale and four times that of a human fingernail (Fig. 3B). Additionally, the epidermal thickness of the elephant trunk is nearly 100 times thicker than the epidermis on an adult human's torso.

The next skin layer beneath the VE is the dermis. We observe two regions of increased thickness of the dermis, the tip and the proximal base. At the tip, the ventral dermis is 1.5 thicker than the dorsal dermis (2.3 mm

versus 1.46 mm thickness, respectively) (Table 1 and Supplementary Fig. 3). This thickening makes sense: at the tip, the thicker ventral dermis is where the trunk grasps and manipulates objects. The dermis appears to thicken where the trunk increases in diameter as well. At the proximal base, the dermis along the dorsal trunk is 700% thicker than the dermis at the distal tip (5.44 versus 0.8 mm, respectively).

## Micro-structure of elephant skin

To characterize compositional differences in COL1 between the skin samples from the elephant trunk, we used SHG imaging. SHG can identify the macro and micro-level structures of the skin, such as COL1 fiber density, intensity, and orientation (Supplementary Fig. 4A). The color intensity in SHG images can be used as a proxy for either fiber thickness or the pretension in the fiber[10,11]. Each of these individually indicates the mechanical state of the tissue as thicker fibers and more pretension both correlate to stiffer tissue[10,11]. In analyzing the images taken with linearly polarized light, we found the ventral trunk has an overall higher intensity than the dorsal trunk (Fig. 4A, B), suggesting ventral fibers have more pretension (or ability to resist deformation) or thickness than dorsal (Fig. 4C). At the tip of the trunk, the ventral skin has an SHG intensity twice that ($p < 0.001$) of dorsal skin (Fig. 4D). This trend was accentuated at the trunk base, where the ventral skin SHG intensity was six times ($p < 0.001$) the intensity of the dorsal skin (Fig. 4D). The differences in SHG intensity observed indicate that dorsal skin has less pretension imposed on the collagen, or smaller fibers, allowing more stretchability than ventral skin.

We next used the SHG images to assess the collagen fiber angle (Supplementary Fig. 4 and Fig. 5A). Two fiber angle orientations, perpendicular and parallel relative to the skin surface, are of particular relevance to the physical properties of the skin (Fig. 5B). As discussed in the methods, we define zero degrees as the outward normal of the skin surface (Fig. 5A). Perpendicular fibers resist axial trunk loading from forces perpendicular to the skin (Fig. 5B). The parallel fibers are oriented 90 degrees to the outward normal. Parallel fibers primarily assist with extension and shear loading tolerance (Fig. 5B).

Upon analysis of the collagen orientation from the SHG images, we found that dorsal skin samples are composed of bimodal orientation peaks, with COL1 fibers oriented in both the perpendicular and parallel directions (Fig. 5C). All samples of dorsal skin analyzed have over 20% of perpendicular and 20% of parallel fibers in the skin, indicating a bi-model peak of

**Table 1 | Table displaying the thickness of each skin layer in mm displayed in mean ± standard deviation**

| Distance from tip | SC (mm) | VE (mm) | D (mm) |
|---|---|---|---|
| **Dorsal surface** | | | |
| 3 | 0.39 ± 0.11 | 0.05 ± 0.05 | 0.8 ± 0.097 |
| 27 | 0.17 ± 0.16 | 0.36 ± 0.21 | 1.5 ± 0.27 |
| 81 | 0.39 ± 0.27 | 0.27 ± 0.21 | 6.6 ± 0.28 |
| 100 | 0.87 ± 0.61 | 0.58 ± 0.51 | 5.8 ± 0.90 |
| 133 | 1.8 ± 0.68 | 0.42 ± 0.36 | 5.4 ± 0.64 |
| **Ventral surface** | | | |
| 27 | 0.29 ± 0.22 | 0.50 ± 0.31 | 2.4 ± 0.44 |
| 51 | 0.12 ± 0.18 | 0.32 ± 0.29 | 4.9 ± 0.54 |
| 133 | 0.34 ± 0.22 | 0.22 ± 0.22 | 4.2 ± 0.23 |

Results of each layer are displayed as SC (Fig. 3A), VE (Supplemental Fig. 2), and D (Supplemental Fig. 3).

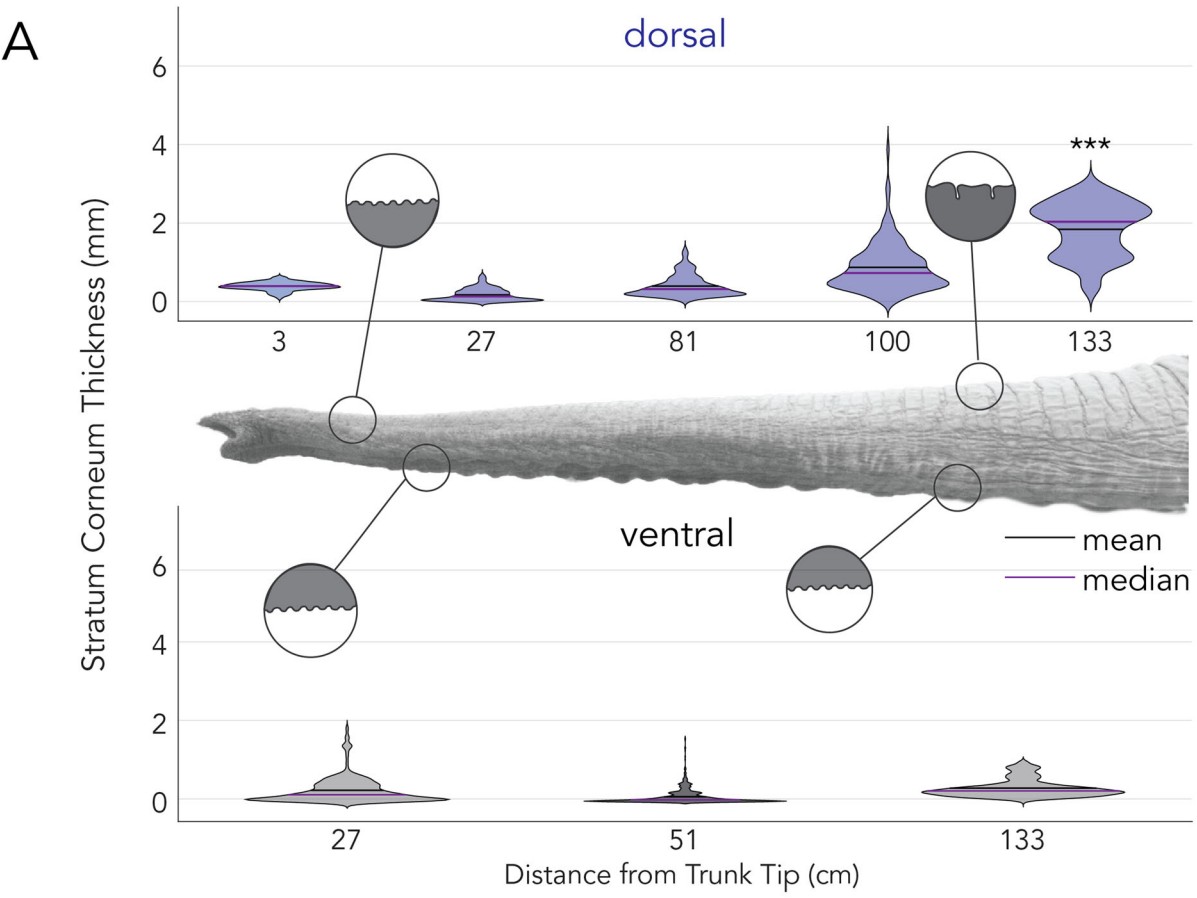

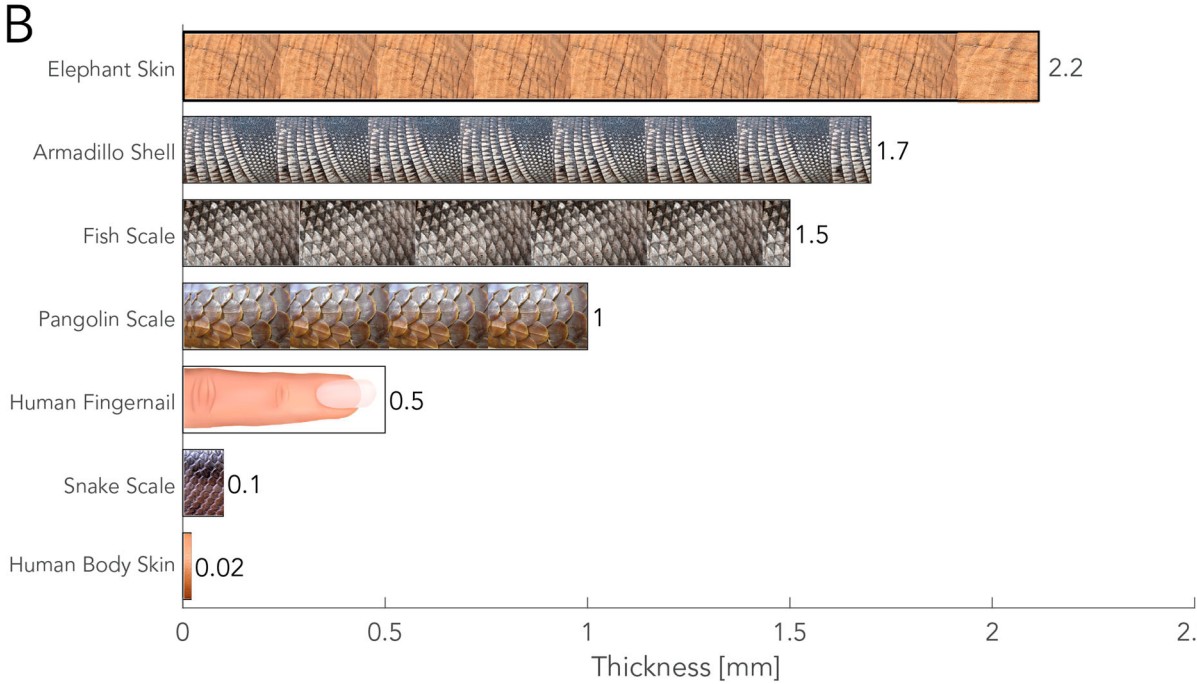

**Fig. 3 | Elephant skin armor thickness compared to other species. A** Relationship between stratum corneum (SC) thickness and position on the trunk. Stars indicate the statistical significance of the difference between dorsal and ventral sites: (***$p < 0.001$). **B** Thickness of different dermal armors across species. Non-elephant data taken from (Wollina et al. 2001, Wang et al. 2016, Yang et al. 2019, Chintapalli et al. 2014, Han et al. 2018, Bordoloi et al. 1995)[23,24,32–35]. Images taken from Adobe CC Images.

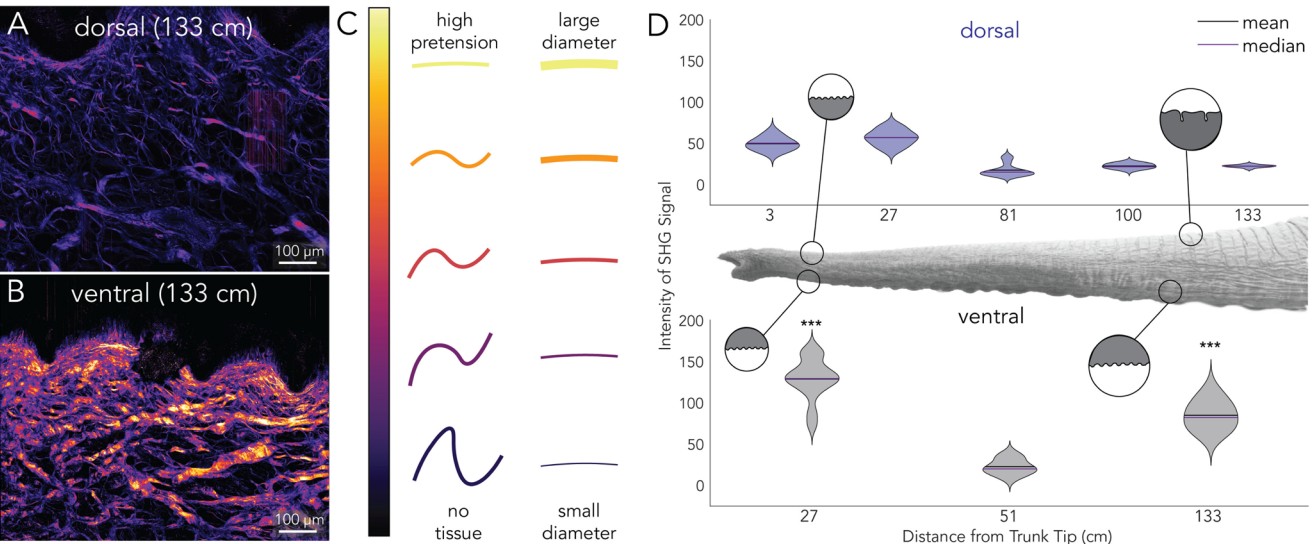

**Fig. 4 | Ventral trunk has higher SHG intensity compared to dorsal trunk.**
**A**, **B** SHG stacked image of dorsal and ventral sections of the proximal trunk.
**C** Schematic displaying the relationship between the intensity of SHG in fibers and the pretension or diameter relationship. **D** Relationship between SHG intensity and position on the trunk with the average of ten regions of interest. Stars indicate the statistical significance of the difference between dorsal and ventral sites: (***$p < 0.001$).

fiber distribution. Additionally, we see a significant difference when we compare the fiber orientation at specific sites along the trunk. Along the dorsal surface of the trunk at 3, 27, and 81 cm from the tip of the trunk, we see significant differences between the percentage of perpendicular and parallel fibers. The proximal base (100 and 133 cm from the tip) on the dorsal surface, however, shows no significant difference, with around 25% perpendicular and 25% parallel fiber orientation (Fig. 5C).

Dorsal and ventral surfaces show statistically significant differences in collagen fiber orientation. At the distal tip of the trunk (27 cm from the tip), the ventral skin has significantly more COL1 fibers in the parallel direction ($p < 0.01$) compared to the dorsal skin at the same site (Fig. 5C). When we look at the proximal base (133 cm from the tip), the dorsal skin has more perpendicular collagen ($p < 0.001$), and less parallel collagen ($p < 0.001$) relative to ventral skin at the same location.

As mentioned above, we see a bimodal distribution of fiber orientation in the elephant with large percentages in both the perpendicular and parallel directions. In our previous work looking at human skin, we found that both plantar (skin on the sole of the foot) and non-plantar (body) skin contained COL1 fibers with preferential fiber orientation (perpendicular or parallel) in just a single direction[22], as opposed to the bi-model distribution observed in elephant skin. Given the differences in fiber orientation between human and elephant skin, we postulated that there would also be differences in the entanglement of COL1 fibers.

Analysis of the COL1 fiber overlap (Fig. 6A) (from two sites in four regions detailed above) revealed that the average number of fiber crossings per µm³ in the dorsal elephant trunk is 5.9 ± 1.3 while in the ventral elephant trunk it is 5.6 ± 1.4 (Fig. 6B). Both of these sites on the elephant trunk have significantly more fiber crossings than found in human plantar (1.1 ± 0.90) and non-plantar (0.85 ± 0.70) skin. Overall, the number of entangled fibers is nearly six times higher in elephant trunk skin compared to human skin.

## Discussion

We set out to evaluate elephant trunk skin architecture along the length of the trunk. We found variations in morphology and composition along the trunk length at both the macro and micro scale. The dorsal portion of the trunk, including the trunk's dorsal finger (3 cm from tip) and dorsal root (133 cm from tip), had the thickest SC layers. The distal tip of the trunk, or finger, is regularly used to manipulate objects, and the dorsal root is more exposed to external stimuli[3]. These functions may explain the thicker dorsal finger and root SC layers. When we combine the thickness of the SC and VE

in this dorsal root and compare it to other species, we see the elephant may have the thickest dermal armor among extant animals; elephants have a dermal armor thickness twice that of a pangolin scale and four times a human thumbnail (Fig. 3B)[23,24].

While the elephant uses skin for protection, aquatic and arctic species rely on thick fat layers for protection and insulation[25]. In humans, the sole also has a fat pad that protects the skeleton from heel strike impact. Unlike the fat layers in arctic species, this fat pad on the heel does not protect the skin—instead, foot skin has adapted to be thicker and stiffer than body skin, which allows it to withstand mechanical loading. In other species, we see a range of morphological structures, such as shells and scales (Fig. 3B), where the skin armor has adapted to provide additional protection against environmental pressures[23].

Our study was limited by having material samples from only one elephant specimen and one elephant species. While many dry skin samples are available in museums, frozen samples, which allow preservation and histological analysis, are much rarer. For some time, the elephant trunk was stored in a −20 °C freezer. This may have impacted the tissue integrity and collagen architecture but would not have impacted skin morphology. Moreover, this specimen was an African bush elephant (*Loxodonta africana*), just one of three elephant species. There may be intrinsic differences between species that we could not address in our study, as skin morphology appears different between African and Asian elephants[26]. Asian elephants have only one "finger" at the tip, with the ventral finger composed of a cartilage bulb. This difference in trunk tip morphology is partly due to Asian elephants being grazers (eating low-lying vegetation). In contrast, African elephants are browsers (eating high-growing vegetation) and require a prehensile finger to grip and pull leaves off branches for nutrients. Another limitation of our study was that we were only able to obtain images through conventional SHG, and not PSHG, using our microscope. The application of PSHG could be used to obtain additional information on skin mechanics, or the ultrastructure of entangled collagen fibers.

Boyle et al. found the skin on different body sites of humans had COL1 fibers oriented preferentially in either a parallel or perpendicular direction, depending on the functional requirements for skin at that site[22]. In contrast, the dorsal surface of the elephant trunk expressed relatively large percentages (≥ 20%) of both perpendicular and parallel collagen. Previous work has shown the mid-dorsal portion of the trunk is more pliable than the mid-ventral skin[2]. Our results show that dorsal and ventral sites have similar bimodal distributions of fibers and similar amounts of collagen

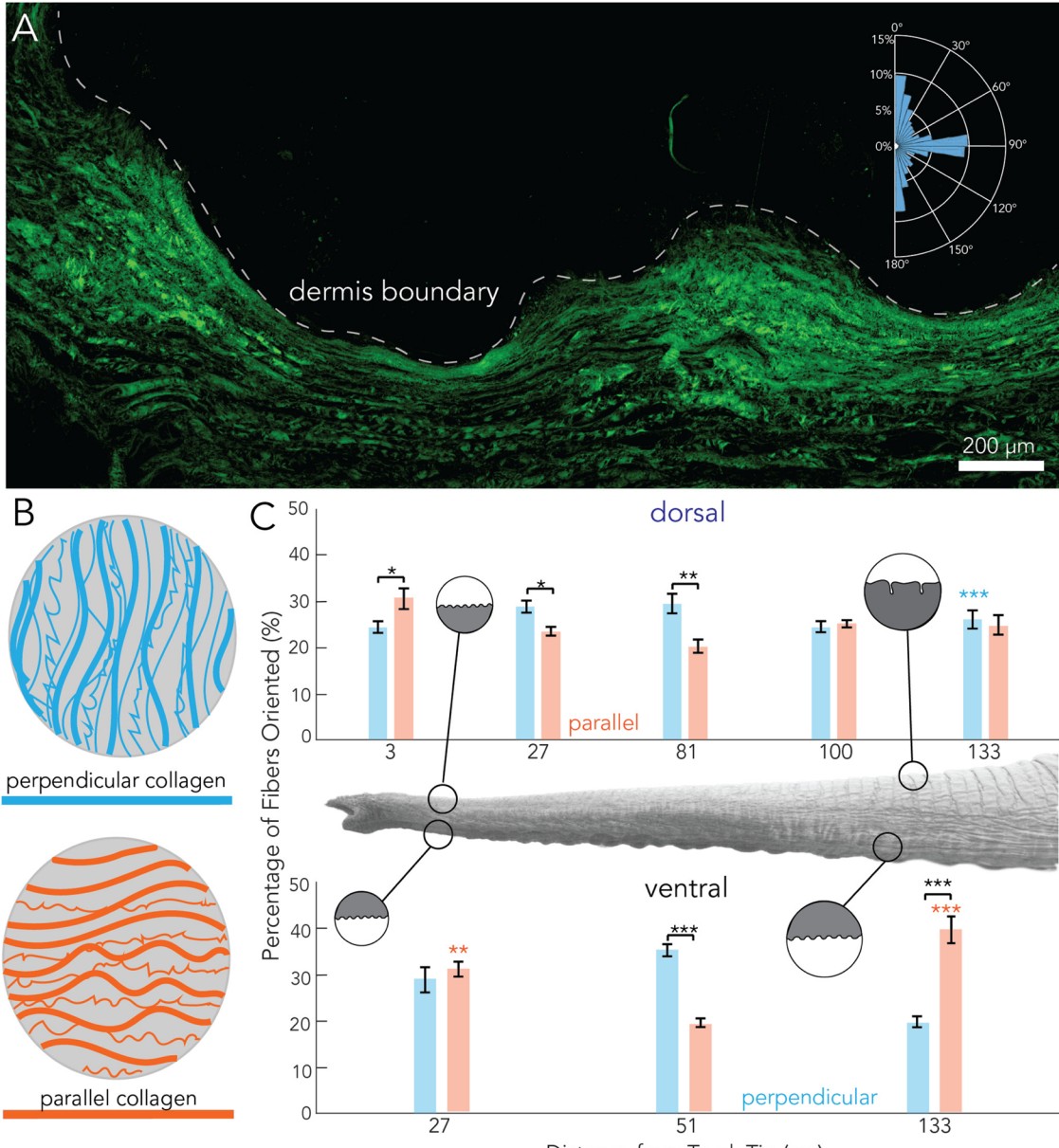

**Fig. 5 | SHG analysis of collagen fiber orientation along the trunk. A** Stacked SHG image of the distal ventral elephant trunk with an inset of CurveAlign histogram output showing collagen fiber alignment of a single ROI from a single image. **B** Schematic of parallel and perpendicular collagen fibers in the dermis. **C** Relationship between the percentage of collagen fibers and position on the trunk. Parallel fibers are shown in orange and perpendicular fibers in blue. Blue and orange stars indicate statistical significance between dorsal and ventral sites, with stars placed over the larger value. Black stars indicate statistical significance between perpendicular and parallel comparisons within a single site. Data were averaged across the four ROIs with error bars indicating standard deviation. Stars indicate the following significance: (*$p < 0.05$, **$p < 0.01$, ***$p < 0.001$).

crossings. However, we see SHG intensity differences between the dorsal and ventral skin along the entire trunk length. This could indicate the collagen fibers in the dorsal portion have less pretension and, therefore, are not entangled with each other unless large strain deformations are introduced. Thus, the ventral skin's increased strain stiffening could be because the fibers are oriented in both directions and are already in tension, causing a faster transition to strain-stiffening behavior, leading to a similar conclusion of Boyle et al. in human skin, i.e., morphology and composition play distinct and complementary roles in elephant skin mechanics. This ventral strain-stiffening could be used to allow more controlled gripping with a stiffer skin surface.

We envisage that these observations will give inspiration to future biomimetic studies. While collagen fiber entanglement is still being understood, the general belief is that the structure on the micro scale leads to unique mechanical responses on the macro scale. There has been increased interest in understanding the macro physical properties that stem from micro-scale entanglements as such work may influence the design of soft robotic manipulators[27]. Our studies of the impacts of woven fiber structure inside the skin are reminiscent of the impact of patterning in knitted fabric structures. Knitting is a centuries-old activity that involves manipulating a string-like material, traditionally yarn, into a complex fabric with emergent elasticity. These fabrics can exhibit vastly different mechanical properties based on how the stitches, specific slip-knots formed by the yarn, are patterned and structured[28]. These structural differences leading to robustness are also challenges in the public health sector. Collagen fibers in skin constructs are primarily oriented

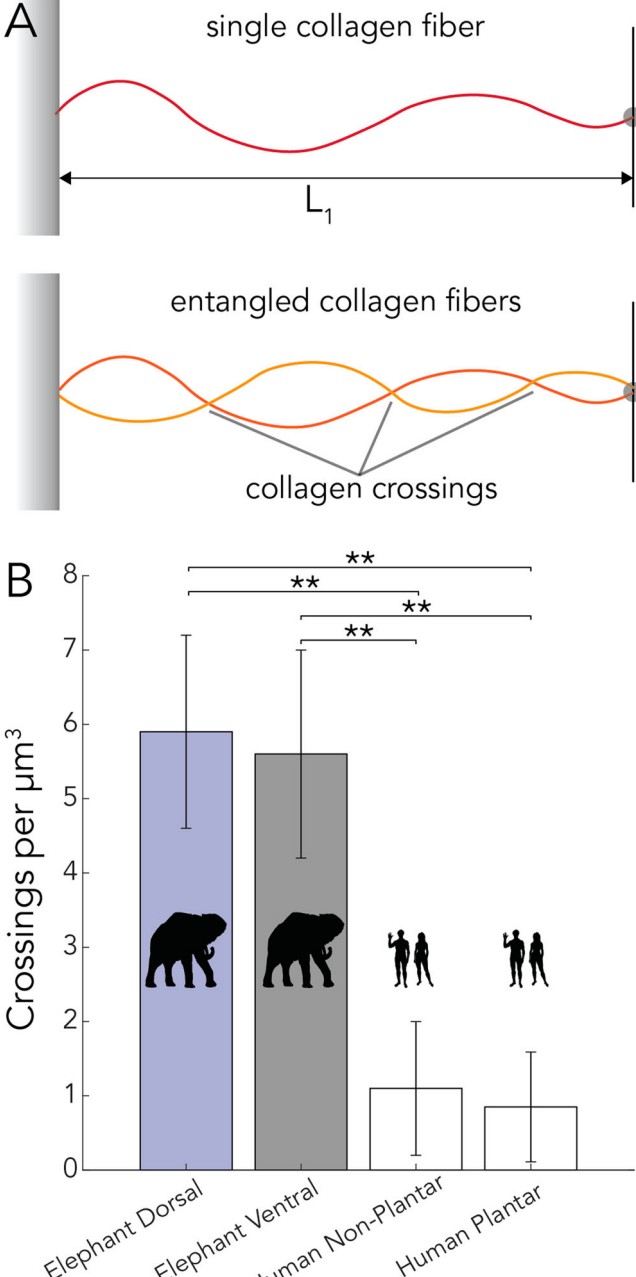

**Fig. 6 | Elephant trunk skin has larger amounts of collagen crossings than human skin. A** Schematic of collagen with crossings and without crossings. **B** Collagen crossings per cubic micron for two elephant skin sites (dorsal and ventral regions 133 cm from the tip) and human plantar and non-plantar skin published SHG images of human skin reanalyzed from ref. 22. Stars indicate the statistical significance of the difference between elephant and human skin: (**$p < 0.01$). Standard deviation is the deviation from the analyzed regions of interest. Silhouettes of African elephant (*Loxodonta africana*) from phylopic artist Agnello Picorelli. Human silhouettes from phylopic.

parallel to the skin dermis as they govern how skin contracts. Orienting perpendicular fiber alignment could make skin grafts more robust in terms of their mechanical and flexibility utility.

In summary, we compared the trunk along the distal-proximal and dorsal-ventral anatomical axes, finding differences in the morphology and composition across the elephant trunk and giving insights into the form-function relationships. Elephant trunks have some of the thickest dermal armor in the animal kingdom, with a 2.2 mm thick epidermis. This armor is

paired with parallel and perpendicular collagen in the dermis, giving it both strength and flexibility. Furthermore, the bi-model orientation of collagen in the dermis leads to individual fiber overlap and interaction, showcasing the entanglement of fibers inside the skin. This work shows the complex nature of elephant skin and provides bio-inspiration for materials that require strength and flexibility.

## Methods
### Dissection of elephant trunk skin
Icahn School of Medicine at Mount Sinai, New York, provided access to a dissected frozen trunk from a 38-year-old female African elephant (*Loxodonta africana*) that initially lived in a Virginia zoo. The elephant was euthanized for health issues in 2011.

We accessed the trunk when it was on loan from the National Museum of Natural History (NMNH), Smithsonian Institution. The elephant's body weight before death was ~4000 kg. The trunk was cut into several parts and initially stored in a freezer at −20 °C until it was dissected in July 2016.

In March 2019, eight samples of the trunk skin were further dissected at the Icahn School of Medicine at Mount Sinai. These samples included five dorsal and three ventral samples ranging from the proximal to the distal end of the trunk. These samples were shipped on dry ice to Imperial College London by the Smithsonian Institute Collections Department as a scientific exchange between the two CITES-registered institutions. The Animal Plant and Health Agency in the UK (authorization number ITIMP19.0822) approved the tissue shipment. The samples were stored at Imperial College London at −80 °C until embedding, sectioning, and imaging were conducted from January to March 2020.

### Histology and morphometrics
The eight samples were further dissected to enable analysis in the trunk's longitudinal direction. Samples were embedded in an OCT (optimum cutting temperature) medium, and 20 μm-thick sections were cut on a cryostat (Supplementary Fig. 1). The tissue sections were stained using hematoxylin and eosin (H&E) and then imaged on a Zeiss inverted microscope at 3x magnification. Images were automatically segmented using the wand tool in FIJI (ImageJ) based on the stained color differences from H&E.

To quantify the thicknesses of each layer (the stratum corneum (SC), the viable epidermis (VE), and dermis (D) shown in Fig. 2), a MATLAB script was used to divide each H&E image (1000 pixels wide) into vertical strips of one-pixel width. The pixels corresponding to each layer were counted and recorded. To compare samples, we reported the thickness for each layer in mm (Table 1, Fig. 3A, and Supplementary Figs. 2, 3).

### Second-harmonic generation
Samples embedded in OCT were cryo-sectioned at 100-μm thickness for second-harmonic generation (SHG) imaging. Samples were imaged on an upright Leica SP5 point scanning confocal microscope, with an HCX IRAPO L 25.0x/0.95NA objective and a zoom of 1 and 512 × 512 pixels in the frame, giving an image size of 620 × 620 microns and xy pixel size of 1.21 microns. For z stacks, the z step was 10.62 microns. Second harmonic imaging was done with a Newport Spectra-Physics Mai Tai DeepSee multi-photon laser tuned to 830 nm (output 2.60 W, horizontally polarized light), pinhole opened to 600 microns, and emission was collected on HyD detector tuned to 400–420 nm with a line average of 3. Tissue regions were imaged as tiles, and images were stitched in LAS-AF (Leica Microsystems). Other channels were imaged with appropriate excitation lasers and HyD detectors with similar settings. All images were taken with linearly polarized light.

Stacks were then processed using a workflow in Fiji (ImageJ), including setting the minimum-maximum range to (0,4000), applying the blur filter ($\sigma = 0.5$), and subtracting the background (rolling ball radius, 40 pixels). TetraSpecks beads (170 nm, excitable with a 488 nm laser) were imaged with the DeepSee multi-photon tuned to 960 nm and the 25x objective, with open pinhole, zoom of five, and 2048 × 2048 pixels in the frame, giving an image size of 124 microns and a xy pixel size of 60 nm, with a z step of 294 nm for stacks, with four line averages. MP laser was set to 960 nm, with an output of

2.58 W, and emitted fluorescence was collected with a HyD detector set between 499–576 nm. Point spread functions from the beads were obtained with Huygens software, giving a Full Width at Half Maximum of 315 nm in x, 300 nm in y, and 1365 nm in z. Total image height varied from 1500–3500 μm, and image length varied from 4500–5500 μm depending on the size of the sliced tissue. To analyze SHG intensity, ten regions of interest of size 200 μm × 250 μm with 150 pixels distance from the boundary were taken and averaged. Completed images are shown in Fig. 4A, B.

**Collagen fiber orientation and crossings**. We used the open-source software CurveAlign to quantify the collagen fiber orientation in SHG images[29]. Images were broken into four regions of interest (ROIs) of size 600 μm × 450 μm with at least a 150 pixel distance from the boundary (Fig. 5A). Once ROIs were acquired, the image stacks were then separated into signal images. For this study, we only examined individual fibers instead of the entire fiber network. To allow comparison of the elephant tissue to human tissue, we used the same settings in CurveAlign as Boyle et al., keeping a fraction of coefficient of 0.06[22] and a distance from the boundary to evaluate as 150 pixels.

We considered two broad categories of fiber orientation shown in Fig. 5B. Fibers perpendicular to the skin, shown in blue in the schematic, have angles of 0 ± 5° and 180 ± 5°, where 0° is defined as outward normal from the skin, as shown in the inset of Fig. 5A. Parallel fibers (orange) have angles of 90 ± 5°. We report the percentage of fibers oriented in each direction. A histogram of fiber arrangement is constructed and analyzed for the perpendicular and parallel orientations (Fig. 5A, inset).

We measured the number of collagen fiber crossings from the dorsal and ventral sections 133 cm from the tip. A region of interest consisted of a 200 × 200-pixel square and an extruded depth of 100 μm. In total, four regions of interest were analyzed. These crossings were counted using Fiji (ImageJ)[30]. In reporting individual collagen fibers, we compared the SHG images of human skin given by Boyle et al. with that of the elephant skin samples in our study[22]. We measured the average number of overlaps per unit volume and compared this with human plantar and non-plantar tissue.

## Statistical methods

All calculations, including statistical analysis, were performed with MATLAB 2022A. In the tables and on the figures, values are reported as mean plus or minus standard deviation. In all data cases, the Shapiro–Wilk test was used to determine normality, and Levene's test was used to assess the homogeneity of variances.

We used the MATLAB function *ttest* for *t*-test to find statistically significant differences between dorsal versus ventral values, difference of values at different positions along the trunk, and differences in perpendicular versus parallel values.

## Reporting summary

Further information on research design is available in the Nature Portfolio Reporting Summary linked to this article.

## Data availability

Data for the manuscript, including the raw SHG images, plotted data points, raw intensity readings, thickness measurements, and sample information, is included on Dryad[31].

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

## Acknowledgements

Thank you to the European Hair Research Society, who supported the travel of AS to Imperial College London. We thank J. Ososky and the Smithsonian Institution Museum of Natural History for assistance with information regarding the frozen elephant trunk. We thank the Facility for Imaging by Light Microscopy (FLIM) at Imperial College London, which is in part, supported by funding from the Wellcome Trust (104931/Z/14/Z) and BBSRC (BB/L015129/1). CH was funded by a project grant from the Engineering and Physical Sciences Research Council (EP/N026845/1).

## Author contributions

A.K.S., D.L.H., and C.A.H. conceived the idea and designed the experiments. J.S.R., D.L.H., and C.A.H. supervised the research project. A.K.S., M.P., S.S., D.C.A.G., M.B., K.S., and J.S.R. conducted the experiments. A.K.S., M.P., S.S., D.C.A.G., M.B., K.S., and J.S.R. analyzed the data. All authors contributed to the writing and editing of the manuscript.

## Competing interests

The authors declare no competing interests.
