## [Transparent Peer Review file · Communications Biology]

Second Harmonic Generation Imaging Reveals Entanglement of Collagen Fibers in the Elephant Trunk Skin Dermis

Corresponding Author: Dr Claire Higgins

Version 0:

Reviewer comments:

Reviewer #1

(Remarks to the Author)

This paper demonstrates the use of second harmonic generation microscopy for quantifying collagen fiber distribution in elephant skin. The use of elephant skin as a sample is new and may be quite applicable for developing biomimetic devices. It is unfortunate that only one elephant could be used for this study. The paper is well written, but there are comments that need to be addressed before publication.

Major Comments:

1. In Table 1, under ventral thickness, the standard deviation on SC is larger than the SC measurement (e.g. 0.29 ± 0.32). How can this be and what does it mean? Also, it might look more clear to express the error on the thicknesses as one number (e.g. 0.39 ± 0.11 is 0.4 ± 0.1).
2. On line 85, it is indicated that the trunk was stored at -20°C for what seems to be ~ 5 years. I thought tissue is usually preserved at -80°C . Is there concerns with the tissue degrading at -20°C for so many years?
3. Are SHG images collected using transmission geometry or backwards geometry? The sample slides seem quite thick for SHG ($100\ \mu\text{m}$). Is birefringence a concern with fiber orientation and overlap analyses?
4. What is the lateral and axial resolution of the SHG microscope?
5. How are the large SHG images (such as shown in Figure 5A) obtained? Are they stitched from smaller images and if so, how is the stitching achieved?
6. How do you count the number of overlaps per unit volume in an SHG stack? If two collagen fibers overlap at a certain angle, can this lead to SHG cancellation which would skew these values? Furthermore, does the SHG signal decrease as you go deeper? If so, how do you ensure that you are able to count all the overlaps if the SHG signal decreases?
7. How are the standard deviations in Figure 5C and 6B obtained? Is it related to the number of sample areas within a single sample?
8. On line 179-180, it is mentioned that the SHG intensity of the ventral trunk is higher than the dorsal trunk. How was this determined? Circularly polarized light should be used however, it was never mentioned in the methods section or mentioned here.
9. On line 184, it is concluded that dorsal skin has less pre-strain. Are there papers using other techniques that can confirm these observations? It is unclear to me whether more literature exists on studying collagen structure in elephant skin.
10. In general, are there differences between collagen fiber structure in elephants versus humans? Do they give the same SHG intensities?
11. On line 200, it is mentioned that the proximal base on the dorsal surface shows no significant difference in fiber orientation. Why might this be the case? This could be addressed in the discussion section. In general, I found that the discussion did not address why certain differences do or do not exist between fiber orientations at different positions.
12. There are other methods that can be used to obtain quantitative collagen structural data using SHG microscopy such as polarization-sensitive techniques and other texture analysis techniques that may be useful to mention to the reader along with why the techniques used here were chosen.

Minor Comments:

1. Table 1 has two decimals for 0.29 ± 0.32
2. Figures 1, 2, S1 and S4 are missing scale bars.
3. Line 196 and 213 uses the word "bi-model", do you mean bimodal instead?

Reviewer #2

(Remarks to the Author)

The paper entitled, 'Second Harmonic Generation Imaging Reveals Entanglement of Collagen Fibers in the Elephant Trunk

Skin Dermis' highlights the form-function relationships of collagen organization in the skin of a single African elephant. The paper leverages both traditional tissue preparation techniques including H&E staining as well as Second Harmonic Generation (SHG) imaging microscopy, an advanced, label-free, collagen-specific method. Collectively, the results indicate that there are unique structures within the elephant trunk that may be related to structure/function properties of the African trunk. The authors hypothesize that the complex design of the elephant trunk skin may inspire new materials that combine both strength and flexibility. This paper is certainly interesting; however, there are several improvements that are required for acceptance.

1. The description of Second Harmonic Generation imaging instrumentation needs to include several more details. At a minimum, the following information needs to be included: 1) Excitation wavelength, 2) Emission wavelength and filter cubes to select for this wavelength, 3) Microscope objective, what is the magnification, numerical aperture, vendor? 4) Detection, what sort of detectors were used, what was the pixel dwell time, 5) Laser power incident on the sample, 6) verification of circularly polarized excitation light. These details are critical for reproducibility.

2. The image processing steps are also a bit unclear. The authors indicate that after pre-processing steps (blur filter and background subtraction), they applied a machine learning algorithm, iLastik to analyze the difference between the fibers and the background. It is unclear why iLastik was needed to do this after the pre-processing steps mentioned. Furthermore, the use of individual fibers from the iLastik processed images do not seem to be used again.

3. To quantify collagen fiber orientation and crossings, the authors indicate that they used CurveAlign. It is unclear if CurveAlign was launched on the previously processed images (i.e. blur, background subtraction, and iLastik) or if this was on the raw, .tif, image files. The authors indicate that CurveAlign was used to examine individual fibers; however, it is my understanding that CurveAlign is a network based analysis routine and the authors may want to use CT-FIRE instead. I understand that CT-FIRE uses both a Curvelet Algorithm and a fiber-tracing extension algorithm to extract information on single collagen fibers. Furthermore, it is also unclear how exactly the number of collagen crossings were assessed using CurveAlign and it is also strange that only the dorsal section located 133 cm from the tip was considered in this analysis rather than all sampled skin trunk locations. What is the justification for this decision?

4. The authors indicate that the color intensity of SHG can be used as a proxy for fiber strain based upon reference 10. In the cited work, the authors validated that they had circularly polarized light and also placed the tissue of interest (arteries) under active strain to elicit the changes in both collagen fiber structure and SHG intensity. While this is certainly a pertinent paper to cite, the authors need to address the following two points to be sure that this paper can support their claims: 1) validate that you are exciting your tissues with circularly polarized light, 2) slice the tissues in a manner that you can apply strain and confirm that the changes in collagen SHG intensity are in fact related to the amount of initial pre-strain in the tissue at the various sites.

Version 1:

Reviewer comments:

Reviewer #1

(Remarks to the Author)

Reviewer #2

(Remarks to the Author)

Thank you for the revisions on this manuscript, the additional information regarding the sample storage, microscopy configurations are well-received. There are a few concerns that still persist and will be addressed prior to publication:

Major:

1) Figure 6, the authors use the terms cross-linked and non-crosslinked collagen fibers and in the schematic use collagen crossings. To avoid confusion with collagen crosslinking, an alteration of collagen fibers that is typically not resolvable based on collagen fiber images, please consider using only the term collagen crossing to indicate that the physical fibers are intersecting or crossing.

2) Figure 5, the use of CurveAlign and the inset does not appear to have clear identification of collagen fibers with the green lines. The identification of fibers with these lines and their orientation is imperative to make any claims about the perpendicular or parallel collagen fibers. It appears that the CurveAlign parameters may need some adjustment to make the claim.

3) Figure 4, the intensity of collagen fibers is color-coded by the heat map and coordinated with the amount of strain. The authors cite previous papers that support this idea of collagen intensity and strain; however, in this case there seems to be altered collagen fiber thickness. The origin of the SHG signal intensity is due to both the organization and number of permanent dipole moments. How do the authors disentangle the impact of more collagen dipoles from thicker collagen fibers with the amount of strain?

Minor:

- 1) On page 4, CurveAlign is misspelled in the first paragraph line 5 under the Collagen and Fiber Orientation and crossing section.
- 2) On page 5, there is a double parenthesis in the last paragraph line 3, (Table 1, Figure S3).

Reviewer #1:

Major Comments:

1. In Table 1, under ventral thickness, the standard deviation on SC is larger than the SC measurement (e.g. "0.29±0.32"). How can this be and what does it mean? Also, it might look more clear to express the error on the thicknesses as one number (e.g. "0.39±0.11" is "0.4±0.1").

Comment: The table values have been updated so that the standard deviation does not exceed the average measurement. The standard deviation and mean value are essentially equal for that measurement as there are some sections of that tissue with no SC thickness as well as sections with thickness exceeding 0.4. Thank you for pointing out this mistake. However, the standard deviation values for the stratum corneum are quite different, as they change a lot on the skin surface. This could be due to the fact that the skin layers can shift along a wrinkle or fold, giving large standard deviations. All values throughout the manuscript have been double-checked with the raw data.

Change: The table has been updated with the correct measurement and standard deviation. We have also made all values on the table have a standardized two significant digits.

2. On line 85, it is indicated that the trunk was stored at -20°C for what seems to be ~5 years. I thought tissue is usually preserved at -80°C. Is there concerns with the tissue degrading at -20°C for so many years?

Comment: There are concerns with tissue degrading at -20 for so many years. The entire trunk was frozen, and there was not a large enough freezer for the trunk sections to be stored at -80 degrees. The samples, once they were sectioned, were stored at -80. For this reason, with the collagen, we are reporting differences between the dorsal and ventral differences. Also for this reason we could not look at the elastin in the tissue sections and stains.

Change: We have added additional discussion of the limitations of the work as there was possible tissue degrading. Our change is as follows:

For some time, the elephant trunk was stored in a -20 degree Celsius freezer. This may have impacted the tissue integrity and collagen architecture but would not have impacted skin morphology.

3. Are SHG images collected using transmission geometry or backwards geometry? The sample slides seem quite thick for SHG (100 μm). Is birefringence a concern with fiber orientation and overlap analyses?

Comment: SHG images were collected using backward geometry (there is no transmitted detector below the sample on CF3). Therefore, the fiber orientation did not concern birefringence. The overlap analysis allowed the construction of the 3D architecture of the specific fibers, and we did not have a birefringence concern with this either.

Change: We have extensively edited the methodology section to highlight that this is not a concern for the way our microscopy was set up. Several lines were added specifically regarding the second harmonic generation microscopy method, including addressing concerns raised by both reviewers.

4. What is the lateral and axial resolution of the SHG microscope?

Comment & Change: We have added two new paragraphs in the SHG methodology section to highlight all of the specifics for repeatable measurements, including all settings, the xy resolution of 1.21 microns, and the z step of 10.62 microns.

5. How are the large SHG images (such as shown in Figure 5A) obtained? Are they stitched from smaller images and if so, how is the stitching achieved?

Comment & Change: The samples are stitched together from smaller images. We have added information of this into the SHG methods, including, "*Tissue regions were imaged as tiles, and images were stitched in LAS-AF (Leica Microsystems).*" The total size of the SHG images ranged based on the thickness of the sample. Ranges are also included in the methods now, " Total image height varied from 1500-3500 μm and image length varied from 4500-5500 μm depending on the size of the sliced tissue."

6. How do you count the number of overlaps per unit volume in an SHG stack? If two collagen fibers overlap at a certain angle, can this lead to SHG cancellation, which would skew these values?

Comment & Change: The SHG stacks show us the collagen fibers and their orientation. An overlap of the two fibers was considered by looking at the weaving between the fiber matrix. The overlaps were counted using Fiji (ImageJ) with a specific region of interest. As for SHG cancellation, specific angles of fibers could be canceled, but our results for this section were more of a comparison of using the same methodology to compare the number of crossings to that of pseudo-model organism of skin mechanics (humans). We have updated the results and methods of the SHG collagen overlaps to be more clear about the specifics of the technique as well as the comparative nature of the results.

Furthermore, does the SHG signal decrease as you go deeper? If so, how do you ensure that you are able to count all the overlaps if the SHG signal decreases?

Comment: The signal decreases or increases throughout the samples primarily based on the sample site. We also chose small volumetric cubes, 200-200-100 micron cubes. Therefore, for our regions of interest, we are counting overlaps. The slices are 100 microns, so we are counting through the entire slice depth.

7. How are the standard deviations in Figure 5C and 6B obtained? Is it related to the number of sample areas within a single sample?

Comment: These standard deviations are calculated as the difference between the different regions of interest (ROIs). The same number and area of regions were calculated for each (four). We have included the sample numbers for each in the methods but have added them to the legends.

8. On line 179-180, it is mentioned that the SHG intensity of the ventral trunk is higher than the dorsal trunk. How was this determined? Circularly polarized light should be used however, it was never mentioned in the methods section or mentioned here.

Comment: The SpectraPhysics DeepSea multiphoton laser on CF3 uses linearly polarised light. We also averaged across ten regions of interest using intensity on ImageJ and this has also been added to the methods.

Change: We have added a whole section discussing the exact laser parameters, “Samples were imaged on an upright Leica SP5 point scanning confocal microscope, with a HCX IRAPO L 25.0x/0.95NA objective and a zoom of 1 and 512x512 pixels in the frame, giving an image size of 620x620 microns and xy pixel size of 1.21 microns. For z stacks, the z step was 10.62 microns. Second harmonic imaging was done with a Newport Spectra-Physics Mai Tai DeepSee multi-photon laser tuned to 830nm (output 2.60W, horizontally polarised light), pinhole opened to 600 microns, and emission was collected on PMT detector tuned to 400-420nm with a line average of 3.”

9. On line 184, it is concluded that dorsal skin has less pre-strain. Are there papers using other techniques that can confirm these observations? It is unclear to me whether more literature exists on studying collagen structure in elephant skin.

Comment: No other papers have examined the pre-strain on elephant tissue. In general we have changed the wording of this to be more consistent with other literature and instead used pretension. In our manuscript we are examining the specific COL1 protein structure and previous manuscripts have shown the relationship between pretension and SHG signal intensity in COL1, as we discuss in the intro “ The fibrillar structure of COL1 fibers allows the microscopy technique to detect COL1 in the tissue, while the resulting image is related to the amount of pretension on the COL1 fibers.” The manuscript we cite is:

Turcotte et al. 2016 <https://pubmed.ncbi.nlm.nih.gov/26806883/>

Which uses circularly polarized light, however, several manuscripts that use linearly polarized light have come to similar conclusions of pretension, namely

Boyle et al. 2018 <https://www.science.org/doi/10.1126/sciadv.aay0244>

Plotczyk et. al: <https://www.nature.com/articles/s41536-022-00270-3>

The amount of pretension of the COL1 fibers impacts the ability to resist deformation:

Rezakhaniha et al. 2012: <https://link.springer.com/article/10.1007/s10237-011-0325-z>

10. In general, are there differences between collagen fiber structure in elephants versus humans? Do they give the same SHG intensities?

Comment: In general, there are many differences. The intensity of the signals is similar, though if you look at the intensity of the signals in our paper compared to that of humans, you see a different magnitude of intensity between plantar and non-plantar skin. The similarities in collagen fiber structure is much less. Human plantar skin and non-plantar skin have collagen fibers that are oriented in a uni-modal sense, with fibers either being oriented perpendicular or parallel to the outer wall. Elephants have a bi-modal organization of collagen with peaks in both directions, which we believe leads to an increase in the overlap of fibers. We have a paragraph in the discussion comparing that of humans to the elephant.

11. On line 200, it is mentioned that the proximal base on the dorsal surface shows no significant difference in fiber orientation. Why might this be the case? This could be addressed in the discussion section. In general, I found that the discussion did not address why certain differences do or do not exist between fiber orientations at different positions.

Comment: For this publication, we were comparing the elephant sites with the parallel and perpendicular collagen orientations between dorsal and ventral sites. We have not drawn major conclusions comparing the sites as it is a small sample site from each site (n=1) for each site due to difficulty in sample acquisition and preparation for testing. Also there could be differences between male and female skin orientation as well which we need to discuss here. We have added some discussion highlighting connections of our results with that of humans more directly.

Change: The following passage has been added to the discussion:

The dorsal surface of the elephant trunk expressed relatively large percentages ($\geq 20\%$) of both perpendicular and parallel collagen. Previous work has shown the mid-dorsal portion of the trunk is more pliable than the mid-ventral

skin\cite{schulz_skin_2022}. Our results show that dorsal and ventral sites have similar bi-modal distributions of fibers and similar amounts of collagen crossings. However, we see a trend in the SHG intensity difference between the dorsal and ventral skin along the entire trunk length. This could indicate that the collagen fibers in the dorsal portion have less pre-tension and, therefore, are not entangled with each other unless large strain deformations are introduced. Thus, the ventral skin's increased strain stiffening could be because the fibers are oriented in both directions and are already in tension, causing a faster transition to strain-stiffening behavior, leading to a similar conclusion of Boyle et al. in human skin, i.e., morphology and composition play distinct and complementary roles in elephant skin mechanics.

12. There are other methods that can be used to obtain quantitative collagen structural data using SHG microscopy such as polarization-sensitive techniques and other texture analysis techniques that may be useful to mention to the reader along with why the techniques used here were chosen.

Comment & Change: We have added multiple sentences in the introduction highlighting why we chose to use SHG in this manuscript.

Change: The section has been added to the introduction:

SHG is just one technique to understand collagen morphology and composition. Alternative approaches for imaging COL1 include polarized light microscopy (PLM), optical coherence tomography (OCT), ultrasound, and magnetic resonance imaging (MRI).

Minor Comments:

1. Table 1 has two decimals for "0..29±0.32"

Comment & Change: we have updated this and normalized all the significant digits in this table.

2. Figures 1, 2, S1 and S4 are missing scale bars.

Comment & Change: We have added scale bars to all of these figures.

3. Line 196 and 213 uses the word "bi-model", do you mean bimodal instead?

Comment & Change: We have updated this in the manuscript.

Reviewer #2 (Remarks to the Author):

The paper entitled, 'Second Harmonic Generation Imaging Reveals Entanglement of Collagen Fibers in the Elephant Trunk Skin Dermis' highlights the form-function relationships of collagen

organization in the skin of a single African elephant. The paper leverages both traditional tissue preparation techniques including H&E staining as well as Second Harmonic Generation (SHG) imaging microscopy, an advanced, label-free, collagen-specific method. Collectively, the results indicate that there are unique structures within the elephant trunk that may be related to structure/function properties of the African trunk. The authors hypothesize that the complex design of the elephant trunk skin may inspire new materials that combine both strength and flexibility. This paper is certainly interesting; however, there are several improvements that are required for acceptance.

Comment: We agree the methods section in our original submission needed more clarification and we have added explanations to the methods decisions as well as clarification of using linearly polarized light in the manuscript.

1. The description of Second Harmonic Generation imaging instrumentation needs to include several more details. At a minimum, the following information needs to be included: 1) Excitation wavelength, 2) Emission wavelength and filter cubes to select for this wavelength, 3) Microscope objective, what is the magnification, numerical aperture, vendor? 4) Detection, what sort of detectors were used, what was the pixel dwell time, 5) Laser power incident on the sample, 6) verification of circularly polarized excitation light. These details are critical for reproducibility.

Comment & Change: We have added specific information regarding your comments to the methods, which is shown below:

“Samples embedded in OCT were cryo-sectioned at $100\mu\text{m}$ thickness for second-harmonic generation (SHG) imaging. Samples were imaged on an upright Leica SP5 point scanning confocal microscope, with a HCX IRAPO L 25.0x/0.95NA objective and a zoom of 1 and 512x512 pixels in the frame, giving an image size of 620x620 microns and xy pixel size of 1.21 microns. For z stacks, the z step was 10.62 microns. Second harmonic imaging was done with a Newport Spectra-Physics Mai Tai DeepSee multi-photon laser tuned to 830nm (output 2.60W, horizontally polarised light), pinhole opened to 600 microns, and emission was collected on PMT detector tuned to 400-420nm with a line average of 3. Tissue regions were imaged as tiles, and images were stitched in LAS-AF (Leica Microsystems). Other channels were imaged with appropriate excitation lasers and PMT detectors with similar settings. Stacks were then processed using a workflow in Fiji (ImageJ), including setting the minimum-maximum range to (0,4000), applying the blur filter ($\sigma = 0.5$), and subtracting background (rolling ball radius, 40 pixels). TetraSpecks beads (170nm, excitable with 488nm laser) were imaged with the DeepSee multi-photon tuned to 960nm and the 25x objective, with open pinhole, zoom of 5 and 2048x2048 pixels in the frame, giving an image size of 124 microns and a xy pixel size of 60nm, with a z step of 294nm for stacks, with 4 line averages.”

2. The image processing steps are also a bit unclear. The authors indicate that after pre-processing steps (blur filter and background subtraction), they applied a machine learning

algorithm, iLastik to analyze the difference between the fibers and the background. It is unclear why iLastik was needed to do this after the pre-processing steps mentioned. Furthermore, the use of individual fibers from the iLastik processed images do not seem to be used again.

Comment & Change: We agree that the methods in the initial submission were confusing. Ilastik was not done for this manuscript. Analysis of SHG images was threefold, including intensity, fiber orientation, and collagen crossing. Ilastik was not used in this manuscript, and we have removed this mention and updated the full methods based on this and other comments.

3. To quantify collagen fiber orientation and crossings, the authors indicate that they used CurveAlign. It is unclear if CurveAlign was launched on the previously processed images (i.e. blur, background subtraction, and iLastik) or if this was on the raw, .tif, image files. The authors indicate that CurveAlign was used to examine individual fibers; however, it is my understanding that CurveAlign is a network based analysis routine and the authors may want to use CT-FIRE instead. I understand that CT-FIRE uses both a Curvelet Algorithm and a fiber-tracing extension algorithm to extract information on single collagen fibers. Furthermore, it is also unclear how exactly the number of collagen crossings were assessed using CurveAlign and it is also strange that only the dorsal section located 133 cm from the tip was considered in this analysis rather than all sampled skin trunk locations. What is the justification for this decision?

Comment & Change: We have adjusted this portion of the methods to clarify which technique was used for what. Overall CurveAlign was used to quantify collagen orientation. We have included additional information about using CurveAlign and how the sample images were processed. The .tif image files were used after being exported from the microscope and stitched together. We mainly used CurveAlign as this technique was used in human plantar versus non-plantar differentiation, and we wanted to discuss specific differences. We have also updated the section discussing collagen crossings in the manuscript. We agree it is strange that only the dorsal section was included in the collagen crossings portion of this manuscript, and we have added analysis from the ventral section at the same distance.

4. The authors indicate that the color intensity of SHG can be used as a proxy for fiber strain based upon reference 10. In the cited work, the authors validated that they had circularly polarized light and also placed the tissue of interest (arteries) under active strain to elicit the changes in both collagen fiber structure and SHG intensity. While this is certainly a pertinent paper to cite, the authors need to address the following two points to be sure that this paper can support their claims: 1) validate that you are exciting your tissues with circularly polarized light, 2) slice the tissues in a manner that you can apply strain and confirm that the changes in collagen SHG intensity are in fact related to the amount of initial pre-strain in the tissue at the various sites.

Comment: The paper we cite evaluates specific COL1 protein structure, and we are

examining COL1 protein structure in our paper. Although the macro dermal structure in elephants is different, we repeated the same technique used in humans for intensity by:

Boyle et al. 2018 <https://www.science.org/doi/10.1126/sciadv.aay0244>

Plotczyk et. al: <https://www.nature.com/articles/s41536-022-00270-3>

In these papers, the authors use linearly polarized light intensity as a proxy for pretension in the fibers, and the amount of pretension in the COL1 fibers impacts the ability to resist deformation as shown by:

Rezakhaniha et al. 2012: <https://link.springer.com/article/10.1007/s10237-011-0325-z>

In our paper, we only look at COL1 and not at different proteins; therefore, we believe that looking at SHG intensity for COL1 protein as a proxy for pretension and with pretension impacting the mechanical ability to resist deformation is justified. We attempted to observe COLIV and COLIX using our technique but could not get a signal and did not report this here. Our primary conclusion was that the difference in the signal intensity between the dorsal and ventral shows mechanical differences between the tissue.

Change: We have adjusted our wording to shift to pretension instead of pre-strain. We have also updated the wording in the results to suggest pretension instead of directly correlating this.

REVIEWERS' COMMENTS:

Reviewer #2 (Remarks to the Author):

Thank you for the revisions on this manuscript, the additional information regarding the sample storage, microscopy configurations are well-received. There are a few concerns that still persist and will be addressed prior to publication:

Comment: Thank you for your additional comments. We have responded to the major and minor feedback individually.

1) Figure 6, the authors use the terms cross-linked and non-crosslinked collagen fibers and in the schematic use collagen crossings. To avoid confusion with collagen crosslinking, an alteration of collagen fibers that is typically not resolvable based on collagen fiber images, please consider using only the term collagen crossing to indicate that the physical fibers are intersecting or crossing.

Response & Change: The authors agree that this would be highly confusing to an audience. Therefore, we have rephrased the Figure 6A caption to say instead:

Schematic of collagen with crossings and without crossings.

We also removed “cross-linking” as a keyword to avoid readership confusion.

2) Figure 5, the use of CurveAlign and the inset does not appear to have clear identification of collagen fibers with the green lines. The identification of fibers with these lines and their orientation is imperative to make any claims about the perpendicular or parallel collagen fibers. It appears that the CurveAlign parameters may need some adjustment to make the claim.

Response: We agree that the Figure 5 inset needs more precise identification. One of our early additions was that we did not use any regions of interest near the boundary, as CurveAlign misidentified some of the fibers in those images, as shown in our old inset. To alleviate this, we analyzed only ROIs 150 pixels away from the boundary. The data displayed is for ROIs that are all 150 pixels away from the boundaries. The image shown is the representative curvelets used in CFR mode, and we agree that it is confusing and was meant as a methodology representation of CurveAlign.

Change: We removed the inset as it obscures some of the data and instead show an example of an ROI taken in the tissue to show the distance from the boundary (150 pixels) and the exact size of the ROIs analyzed in the tissue. As the reviewer mentioned, it is confusing, making the reader feel the parameters need to change. The parameters we chose were consistent throughout and similar to those of the paper by Boyle et al. (2018). This allowed us to compare our alignment results to previously published human

plantar and non-plantar skin alignment results. The updated Figure is included here, and we have updated the caption to reflect this change.

3) Figure 4, the intensity of collagen fibers is color-coded by the heat map and coordinated with the amount of strain. The authors cite previous papers that support this idea of collagen intensity and strain; however, in this case there seems to be altered collagen fiber thickness. The origin of the SHG signal intensity is due to both the organization and number of permanent dipole moments. How do the authors dis-entangle the impact of more collagen dipoles from thicker collagen fibers with the amount of strain?

Comment: Based on both of the responses to reviewers, we updated it not to say strain but instead what the other papers have indicated, which is pretension in the collagen. Also we agree that we cannot conclude or dis-entangle the collagen brightness as either fiber thickness or pretension. Therefore we have added that the intensity could be from either pretension or fiber thickness, both of which have been shown to increase for COL1 SHG intensity. Both fiber thickness and pretension both cause mechanical differences where prior work has shown that higher thicknesses of collagen leads to higher material

modulus (<https://pmc.ncbi.nlm.nih.gov/articles/PMC7525820/>), which is the same case for pretension. Therefore, our interpretation of the results remains the same with the dorsal having lower intensity, leads to softer tissue compared to the ventral skin with higher intensity, likely indicating either larger fibers or higher pretension. Throughout the manuscript, we have added fiber thickness whenever pretension is mentioned. Furthermore, we have updated Figure 4, which we discuss in the next paragraph and the updated Figure 4 is included below.

Regarding collagen thickness, fibers are very thick in the dorsal and ventral sections. We agree that the SHG signal is due to the fibers' organization and the overall fiber size. We realize that in Figure 4C the inset of the “High Strained” fiber and “No Tissue” could be confusing to the reader as all of the fibers are the same size.

We have removed the “High-Strained ” and fiber examples in the figure and instead replaced them with high pretension.

Minor:

1) On page 4, CurveAlign is misspelled in the first paragraph line 5 under the Collagen and Fiber Orientation and crossing section.

Response & Change: We have updated the manuscript and corrected this misspelling.

2) On page 5, there is a double parenthesis in the last paragraph line 3, (Table 1, Figure S3).

Response & Change: Thank you for this feedback. We have removed the parenthesis.